# Impact of Smart Charging for Consumers in a Real World Pilot †

**Pieter C. Bons [1,\*], Aymeric Buatois [1], Guido Ligthart [1], Frank Geerts [2], Nanda Piersma [1] and Robert van den Hoed [1]**

[1]   Amsterdam University of Applied Sciences, Urban Technology, 1097 DZ Amsterdam, The Netherlands;
      a.buatois@hva.nl (A.B.); Guido.Ligthart@hva.nl (G.L.); n.piersma@hva.nl (N.P.);
      r.van.den.hoed@hva.nl (R.v.d.H.)
[2]   ElaadNL, 6812 AR Arnhem, The Netherlands; frank.geerts@alliander.com
\*    Correspondence: p.c.bons@hva.nl; Tel.: +31-6-2850-4480
†    This paper is an extended version of our paper published in 32nd International Electric Vehicle Symposium
     2019 (EVS 32), Lyon, France, 19–22 May 2019.

**Abstract:** A smart charging profile was implemented on 39 public charging stations in Amsterdam on which the current level available for electric vehicle (EV) charging was limited during peak hours on the electricity grid (07:00–08:30 and 17:00–20:00) and was increased during the rest of the day. The impact of this profile was measured on three indicators: average charging power, amount of transferred energy and share of positively and negatively affected sessions. The results are distinguished for different categories of electric vehicles with different charging characteristics (number of phases and maximum current). The results depend heavily on this categorisation and are a realistic measurement of the impact of smart charging under real world conditions. The average charging power increased as a result of the new profile and a reduction in the amount of transferred energy was detected during the evening hours, causing outstanding demand which was solved at an accelerated rate after limitations were lifted. For the whole population, 4% of the sessions were positively affected (charged a larger volume of energy) and 5% were negatively affected. These numbers are dominated by the large share of plug-in hybrid electric vehicles (PHEVs) in Amsterdam which are technically not able to profit from the higher current levels. For new generation electric vehicles, 14% of the sessions were positively affected and the percentage of negatively affected sessions was 5%.

**Keywords:** smart charging; electric vehicles; energy transition; charging infrastructure

## 1. Introduction

Electric mobility is developing at a rapid pace. Market shares for electric vehicles (EVs) are increasing beyond niche markets particularly in leading countries such as Norway (market share of 46% in 2018), Sweden (8%) and the Netherlands (7%) [1]. Worldwide the market growth has been higher than 30% for five consecutive years, leading to an accumulated number of 5 million EVs on the road by 2019 [1]. The electrification of transport constitutes a considerable additional load for the electricity grid. The impact of electric mobility may amount up to an additional 15% of average annual electricity demand of households [2]. Given that charging profiles of EVs tend to overlap with household consumption profiles, the power consumption peaks will significantly increase as a result of EV charging [3] and the limits of the grid capacity may be reached [4–6]. As such, electric mobility provides a substantial challenge to grid operators to facilitate sufficient capacity for charging EVs, maintain grid stability and security while limiting investments in grid reinforcements.

Charging vehicles in a more controlled way, generally defined as 'smart charging', provides opportunities to reduce the impact of EVs on the grid. Smart charging can be defined as the process where charging of an EV is varied on two dimensions (i) time (e.g., postponed charging) and/or (ii) current levels offered at the charging station (e.g., slower charging). Smart charging is considered in order to optimize the effect on objectives such as (i) reduction of net impact by EVs (by reducing charging levels at peak times), (ii) minimizing energy costs (by charging at moments of low electricity prices on energy markets), or (iii) matching with renewable energy generation (by increasing charging speeds and storing the energy in the vehicle) [4].

In recent years smart charging has been an increased topic of research, initially more focused on modelling and simulating smart charging practices. Simulation models have been applied to analyze effects of smart charging on grid impacts [6–8], energy market prices [9,10] and matching of renewable energy profiles [11]. Typically studies indicate that grid impacts can be significantly reduced during peak moments, the extent to which largely depends on charging profiles and assumptions applied in the models [8,10,12].

Similarly, electricity costs for charging EVs can be reduced in the range of 10–50% by applying smart charging [9], which largely depends on local energy market conditions and the characteristics of the applied smart charging profiles. For instance, applying a relatively straightforward postponement strategy for a charging session will deliver less economic gains compared to a 'cut and divide' strategy where a session is split into short sessions of, e.g., fifteen minutes [13]; the latter having the advantage to match charging and fluctuating energy prices much better. Lastly some studies show how smart charging may be applied to increase the match with renewable energy generation [11,14–16], while incentives may be applied to stimulate users to allow for rescheduled charging using for instance time of use (TOU) tariffs [17].

While simulation work has its value in estimating impact of smart charging strategies, in most cases this remains a theoretical exercise. Studies reporting on the benefits of smart charging include many assumptions on charging behavior based on start time distributions, charge volume distributions, average power level of charging equipment and the potential of rescheduling charging sessions [18–21]. Typically these models lack real-world data on actual charging speed and tend to underestimate factors such as double-occupancy (two vehicles on one charging station) and differences in charging speeds between EV models. Both factors may have a significant effect on actual charging speeds. For instance, double occupancy can lead to significantly lower charging speeds per vehicle, as the available current of a particular charging station must be shared between two EVs. In the Netherlands, where public charging stations with two sockets are dominant, utilization data on more than 4000 public charging stations shows how average occupancy of these stations is more than 30% (in Amsterdam even close to 50%) [22]. During evening times double-occupancy can reach up to 70% [22], making double-occupancy a significant factor for assessing the smart charging potential as well as its grid or consumer impact.

Furthermore, EV models differ significantly in charging speeds, which influences the impact of varying charging currents as is typically applied in smart charging schemes. For instance, EVs that allow high current charging (e.g., 32 Ampere or 32 A) are more affected by smart charging than EV models that are internally restricted to 16 A charging. As such, the composition of the EV fleet in a country is a significant factor in the impact smart charging has on the charged volume as well as on the grid. These factors are generally not considered in above mentioned studies.

As a result, simplified assumptions regarding occupancy and actual charging speeds of EVs may result in erroneous results and over-optimistic opportunities for smart charging. Empirical cases in which actual charging data is analyzed can validate or improve assumptions regarding actual charging speeds of EVs with different smart charging schemes. A detailed analysis of the charging characteristics of EVs as well as double-occupancy should be considered. As such, real-world pilots with well monitored data may improve the quality of assessments on the opportunities and consequences of smart charging in practice.

In recent years a number of studies have presented early results of pilots where smart charging was applied in practice [10,23,24]. Overall these studies confirm the opportunities of smart charging to significantly reduce grid impact and indicate that smart charging can be applied on a majority of sessions while still allowing them to reach a full state of charge. However, the amount of structured evaluations of smart charging pilots is still limited and pilots tend to have a limited amount of chargers [25]. Moreover, consumer effects of smart charging are limitedly dealt with, mostly through surveys amongst pilot participants. A quantitative analysis of how many users or sessions are affected by smart charging is largely lacking. This is particularly relevant as most pilots relate to residential (or private) charging rather than public charging [26,27]. Given that residential charging with one EV is more predicable than public charging (with several EVs with different charging characteristics on one charging station), applying smart charging on public charging infrastructure requires more research. This is particularly so in countries (like the Netherlands) where a significant share of EV drivers do not have their own driveway and are dependent on public infrastructure for charging their EVs. Knowledge is lacking whether smart charging practices can be applied on public charging infrastructure and to what extent EV drivers are affected by these practices.

This paper presents the results of a medium scale smart charging pilot carried out in the period January–September 2018 in the city of Amsterdam, the Netherlands [28]. The main optimization objective of the project was to reduce impact on the electricity grid. This was operationalized by providing a varying charging profile, called Flexpower, in which the charging current was varied during the day according to peak and off-peak hours. The Flexpower profile was created specifically for this project and was applied in this pilot for the first time. The profile is static and pre-determined both in terms of current level as well as time of day. As such, the smart charging profile has a similar effect as a 'time-of-use' EV electricity price with fixed off-peak and peak times, rather than a dynamic program where power levels are varied based on real-time conditions (e.g., such as market prices or grid congestion). Apart from lower current levels during peak-hours in the evening, the Flexpower profile also provides higher than normal current levels during daytime to compensate for the lower levels during peak-hours. This smart charging pilot is unique in that it combines a (i) higher than normal and (ii) lower than normal power level, which can lead to both negatively as well as positively affected sessions (rather than only negatively affected sessions found in other smart charging pilots).

In total the pilot included 39 charging stations which were prepared and equipped with software to provide the flexible charging profile. For this purpose data of actual charging transactions as well as smart meter data on Flexpower stations and reference stations was used to analyze to what extent flexible charging profiles reduced the impact of EV charging on the grid and to what extent it affected the charging volume and charging speed for EV users. A detailed analysis of charging speeds of individual EVs was made while the impact of varying current levels on the actual charging speeds was assessed. Furthermore, double occupancy effects on charging speeds and impact of the Flexpower profile were evaluated.

This paper adds to current literature on smart charging with a real-world case of a static smart charging profile. The case adds to our knowledge on opportunities and barriers of smart charging on public charging infrastructure, where charging sessions are less predictable, with more EV users, and charging profiles are more diverse than on residential chargers. The case adds to our understanding under which circumstances EVs may be integrated in our energy grid in a responsible way, also by considering actual charging levels of EVs and double-occupancy effects. The paper ends with a discussion and several policy implications.

## 2. Materials and Methods

This paper describes an experimental study where the impact of the Flexpower smart charging strategy is evaluated on 39 public charging stations by analyzing the data from the charging transactions and smart meters. The materials and methods section describes how the charging infrastructure had to be modified to be able to implement the time-dependent profile, how the data

was gathered, cleaned and enriched by calculating derived variables and classifications, and how the experiment was designed in a robust way in order to deal with a number of known and unknown factors that influence the charging speed of an EV.

## 2.1. Infrastructure

The existing public charging infrastructure in Amsterdam consists of around 2600 charging points (January 2019, [29]) and is an important facility for EV drivers since most households do not have their own driveway. The average occupancy rate is about 50% but increases to 70% during the night [22]. The infrastructure is built on $3 \times 25$ A connections (three phases with a current limit of 25 Amperes), which constitute the standard connection category for Dutch households. It is financially advantageous for the charging point operators to use this connection category for charging stations since the costs of this type of connection are €252 per year compared to €949 per year for a $3 \times 35$ A connection. All these rates are legally fixed, as well as the delivery time of newly requested connections. The impact of a charging station using such a connection on the electricity grid is, however, very different than that of the average household. Peak loads of up to 17 kW are possible for EV charging, while a Dutch household consumes on average around 1 kW [30]. This potentially has severe implication for the stability and reliability of the grid. A second reason to challenge the current implementation is that many newer and more advanced EVs support charging at higher currents than 25 A, making higher connection categories attractive for end users.

In this pilot a higher charging current of 35 A is offered overall, but with limitations around the morning (20 A) and evening peak hours (6 A). This represents a 'best of both worlds' scenario where EV drivers can profit from higher charging speeds when the grid is underutilized but has a lower contribution to the grid load during periods of high demand. This should benefit EV drivers, the grid operator and the charging point operator. Moreover, it may contribute to a better overlap with solar and wind production. While current market shares of wind (1.7% in 2018 ) and solar (0.6%) in the Netherlands are low [31], the projected growth in the coming years may significantly increase the urgency to apply demand response strategies of EVs to match charging demand with renewable energy generation.

The pilot consists of 39 charging stations with two sockets each running the time dependent charging profile (Figure 1). The profile describes how the current limitation, which is kept the same for each of the three phases, changes over the course of a day. The profile is roughly the inverse of the average consumption pattern of Dutch households which has a small peak in the morning and a large peak during the evening hours [32]. The Flexpower profile can as such be interpreted as the remaining capacity of the grid available for EV charging. A large amount of power is available during periods of low household demand (during the night and the middle of the day), but there is less power available during the morning and evening peak consumption periods.

The charging behavior of consumers is very different during weekdays compared to the weekend, as can be seen in Figure 2. The evening peak is lower and starts earlier during the weekend and there is no morning peak. These strong differences make it difficult to interpret aggregated results over the whole week. As a result, weekends were discarded as they have a lower peak load and better overlap with solar power generation and therefore pose less of a problem for the grid than weekday sessions. The results in this paper are only based on data on weekdays (Monday–Friday).

Several technical modifications had to be performed in order to make the existing charging stations in Amsterdam suitable for the smart charging profiles that exceed 25 Amperes. The connections were upgraded from the $3 \times 25$ A category to the $3 \times 35$ A category. The extra costs associated with this upgrade were sponsored by the municipality of Amsterdam. In time the results of this pilot may help to introduce a new rate in the legislation for a flexible connection ($3 \times 35$ A with limitations). The firmware of the charging stations has been updated from OCPP (Open Charge Point Protocol) 1.5 to OCPP 1.6 to allow time dependent current limits and remote configuration.

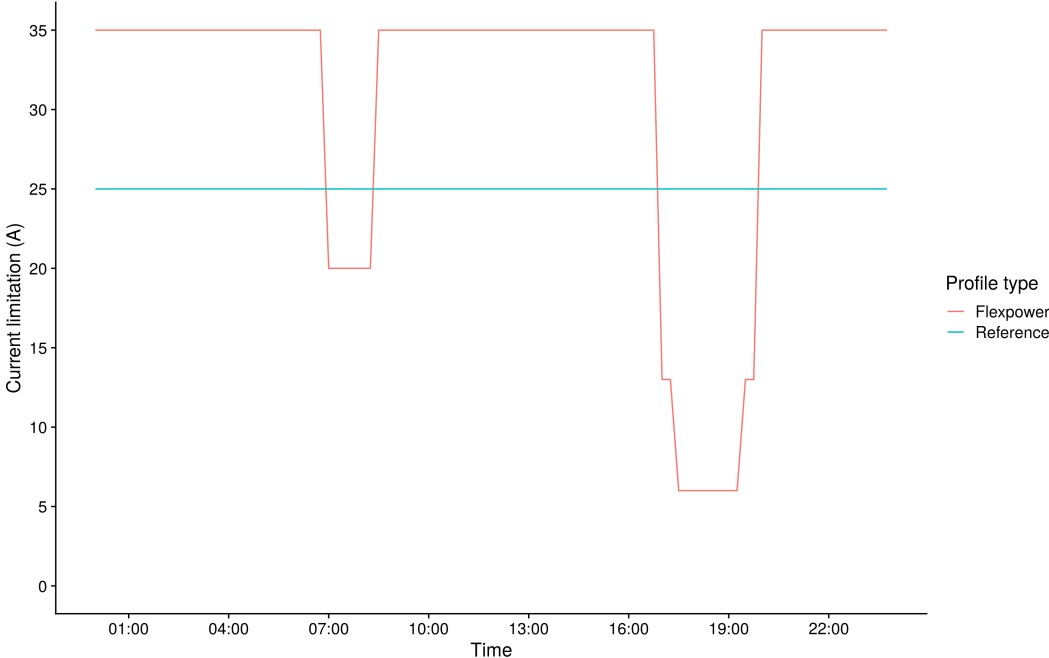

**Figure 1.** The time-dependent current profile deployed on the selected Flexpower charging stations on weekdays compared to the current limitation on a regular public charging station (reference) in Amsterdam.

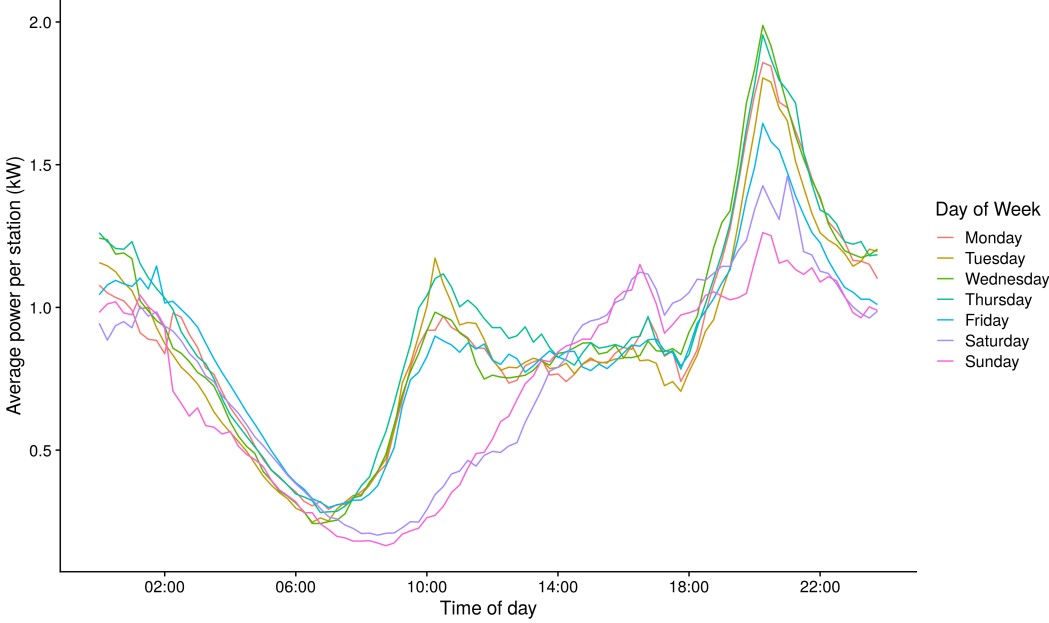

**Figure 2.** The average power per station over the course of the day for each day of the week.

Another modification that was made is that the 16 A fuses which were present on individual sockets were removed, and the distribution of current over the sockets was fully controlled by software. This has the advantage that the full current of the station is available when only one socket is connected, while this was previously limited by the fuse. The charging station model that is used in the pilot is the EVBox PublicLine.

The pilot was communicated through information on the municipal website in which the pilot was explained. Only in phase 2 of the project (started in 2019 and is beyond the scope of this paper) more targeted information to EV drives was provided through information on the charging stations with the

Flexpower profile. As a result, it is expected that most EV drivers using Flexpower stations were not or hardly aware of the Flexpower profile and that rebound behavior, for instance EV drivers deliberately avoiding or favoring Flexpower stations, is not likely to be observed. In phase 2 of Flexpower possible rebound behavior is an object of study.

*2.2. Data*

Data on all transactions on public charging stations in Amsterdam is collected in the CHIEF database (Charging Infrastructure Efficiency database), which is managed and hosted by the Amsterdam University of Applied Sciences [22]. This data contains aggregated information on each charging session, such as the start time, end time, duration and total energy of the transaction as well as the Radio-frequency identification (RFID) of the user (the unique identifier of the contactless payment method such as payment card or smartphone). The data is pushed to the CHIEF database by the charging point operator on a monthly basis. The data set contains personal details and therefore cannot be published.

The transaction data does not have a sufficiently high resolution for the Flexpower analysis, most notably it lacks data on the actual charging time. For example, a transaction with a duration of 4 h and total energy of 44 kWh could have been achieved by non-stop charging at 11 kW or by charging at 22 kW for 2 h followed by two more hours of idle time. To address this issue, more detailed data was collected during the eight-month period from January 2018 to September 2018. This charging data, which we will refer to as metervalues, contains a record for every 15-minute interval during charging and for every 2-hour interval during idle connection time. This allows us to monitor the charging progress during the connection period and the two scenarios as described in the example earlier in this paragraph are now easily distinguishable.

The metervalue data was delivered in CSV format by Nuon/Vattenfall. The metervalues alone are also not sufficient to do the analysis for two reasons: The metervalue data does not contain RFID information, so it would not be possible to connect different transactions by the same user. The second reason is the metervalue data does not cover the full charging session. The first metervalue is recorded 15 min after the start of the transaction and the last meter value is recorded some time before the end. This means that start and end times cannot be matched between the two data sets and also the difference between the last and the first metervalue is often slightly smaller than the total energy found in the transaction data (some energy is charged in the first 15 min and in the last couple of minutes).

The two data sets do not have a shared transaction ID column which can be used for merging. Given that the data sets do not have an overlapping start time, end time or total energy there is no single unique property to use as a match between the two data sets. For this reason, a rolling merge was performed by finding the transaction that has a start time before the first metervalue of the session and an end time after the last metervalue on the corresponding connector and charging station. This gave a 99% match. The difference can be explained by the removal of several records in the data cleaning stages implemented in the CHIEF database.

All analyses were performed using the statistical programming language R (version 3.3.3).

*2.3. Data Cleaning*

A filtering process was required to remove several erroneous data records. The filtering consists of the following steps:

- Records with only one metervalue for the whole transaction are removed as it is not possible to compute the power (which requires taking the difference between multiple energy values).
- Records with power higher than 22 kW are removed.
- Similarly, records with "negative" charging (where the metervalue decreases over time) are removed.

- Transactions with zero total energy are removed (possibly indicating a failure in the communication between vehicle and charging station).
- Transactions charging over 100 kWh are removed, given that the largest battery size available is 100 kWh.
- Transactions without matching RFID were discarded.

The count of the filtered transactions is shown in Table 1. In total about 9% of the transactions were removed from the data set, leaving 43,904 charging transactions for analysis.

**Table 1.** Overview of removed data records.

| Numbers of Transactions | Removed Transactions | Explanation |
|---|---|---|
| 48,152 | - | Raw number of transactions |
| 45,657 | 2495 | Transactions with only one record (or multiple transactions but very close together so effectively only one) |
| 45,620 | 37 | Transactions with charging > 50 kW or negative charging |
| 44,326 | 1294 | Transaction that do not charge at all (total energy = 0) |
| 44,324 | 2 | Transactions charging > 100 kWh |
| 43,904 | 420 | No RFID match from transactions data |

*2.4. Experimental Design*

During the operational pilot multiple Key Performance Indicators (KPIs) were monitored to evaluate the effects of the smart charging profiles. The KPIs were derived from discussions with relevant stakeholders (municipality, grid operator, charge point operator) in which the most prominent indicators for success of the project were identified:

1. Effective charging speed (kW) as a function of time of day,
2. Amount of charged energy (kWh) per charging socket as a function of time of day,
3. Number of positively/negatively affected sessions in terms of charged energy per transaction.

The KPIs reflect the particular interests of the stakeholders such as grid operators (effective charging speed or charging demand in kW), charge point operators (amount of energy charged per day) and municipalities (to what extent are EV-drivers affected by the Flexpower profile). Other indicators that were identified but not considered in this study are the "match between charging demand and renewable energy generation" as an indicator for "green charging" and "percentage of power reduction during peak times" indicating actual contributions to peak power reduction. For both indicators detailed data (on local renewable energy generation and data on power grids, respectively) was required that was not available during the time of study. These indicators will be considered in future work.

The KPIs can be calculated from the combined data sets (transaction data and metervalues). To be able to determine the actual impact of the smart charging profiles, 52 different charging stations were selected as a reference group. The reference stations were not modified and provided a static current level to EVs. They use the standard $3 \times 25$ A connection with two sockets, which each have a 16 A fuse limiting the current to 16 A for a given connected vehicle. Given that Flexpower pilot stations were selected by the municipality based on their suitability for Flexpower (high occupancy rates and energy volumes) rather than by using random sampling, it was necessary to verify whether the reference stations, also selected by the municipality, indeed provide a fair comparison group. The two groups were compared on the characteristics (i) energy/session, (ii) connection duration, (iii) start time and (iv) sessions/month to establish whether they have similar distributions. If for example the reference

group would serve vehicles with larger batteries or are located in busier areas, they would give a higher average amount of energy or have higher occupancy and the analysis could give biased results.

The distributions of the different characteristics are presented in Figure 3 and an overview of the summary statistics is given in Table 2. The results show how on average sessions on Flexpower charging stations tend to last slightly longer (10%). This may partly be explained by a slightly higher percentage of sessions starting at evening times at Flexpower charging stations, where reference stations have a slightly higher occurrence of day sessions (with corresponding shorter connection times). Energy transfer is slightly higher for Flexpower stations (5%), which can similarly be explained by the longer connection times. Reference charging stations tend to have slightly higher utilization frequency (8%).

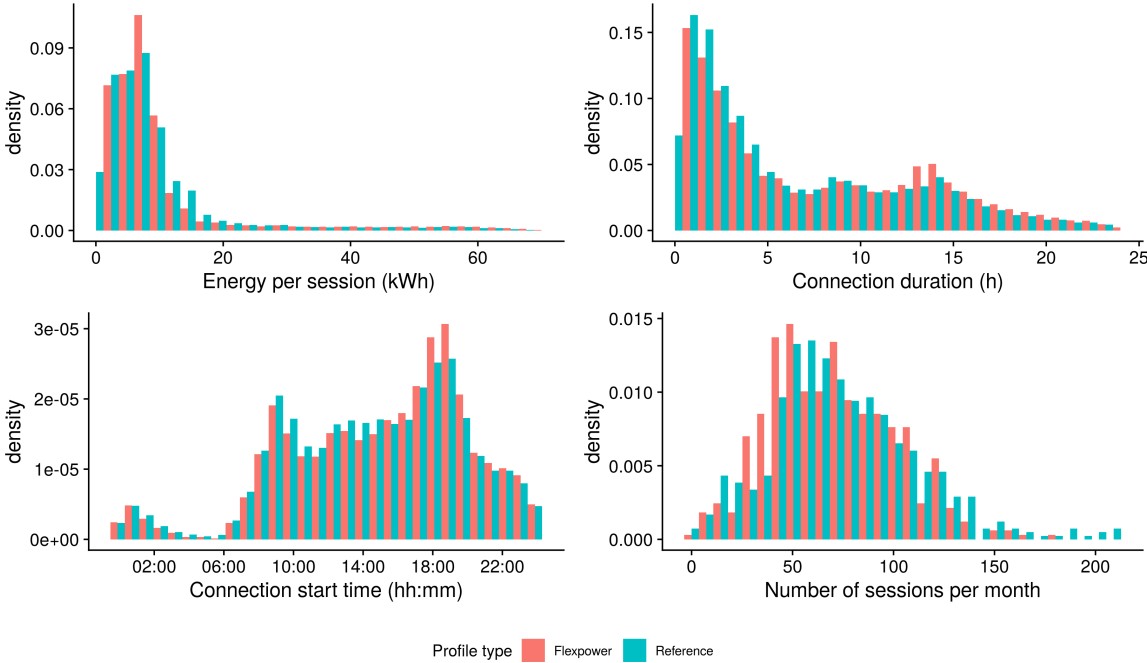

**Figure 3.** Comparison of the Flexpower charging stations with the reference stations on four selected indicators.

All in all, these results show that the two groups have sufficiently similar distributions for fair comparison, both in terms of mean values on the chosen indicators as well as in the overall shape of the distributions, introducing an uncertainty of less than 10%.

**Table 2.** Descriptive statistics of four indicators for Flexpower and reference stations.

| KPI | Flexpower | Reference | Difference |
| --- | --- | --- | --- |
| Energy per session - mean (kWh) | 9.91 | 9.47 | 4.4% |
| Energy per session - standard deviation (kWh) | 12.4 | 11.1 | |
| Energy per session - min (kWh) | 0.01 | 0.01 | |
| Energy per session - max (kWh) | 91.5 | 98.6 | |
| Connection Duration - mean (h) | 9.94 | 9.03 | 9.2% |
| Connection Duration - standard deviation (h) | 14.7 | 15.9 | |
| Connection Duration - min (h) | 0.006 | 0.004 | |
| Connection Duration - max (h) | 514 | 620 | |
| Start Time - mean (hh:mm:ss) | 15:02:18 | 14:36:44 | 25:34 |
| Start Time - standard deviation (h) | 5.16 | 5.18 | |
| Number of sessions per month - mean | 70.0 | 75.2 | 6.9% |
| Number of sessions per month - standard deviation | 31.1 | 36.5 | |

## 2.5. Characterization of Vehicle Categories

There are many different EV models on the market with different charging characteristics. The number of phases that a vehicle uses to charge can differ (there are 1-phase, 2-phase and 3-phase models) as well as the maximum current that the vehicle can use (16 A, 25 A and 32 A). This gives nine unique combinations, six of which have at least one corresponding EV model currently on the market (see Table 3).

**Table 3.** An overview of the different categories of charging characteristics available on the market with examples of corresponding popular models. Many models have multiple versions with different specifications meaning some models may occur in more than one category.

| Vehicle Category | Example of Model on the Market |
| --- | --- |
| 1 × 16 A | Mitsubishi Outlander (PHEV), Volvo V60 (PHEV), Volkswagen Golf (PHEV), Volkswagen Passat (PHEV), Nissan Leaf |
| 2 × 16 A | Volkswagen e-Golf |
| 3 × 16 A | BMW i3, Audi e-tron |
| 1 × 32 A | Jaguar I-Pace, Hyundai IONIQ |
| 3 × 25 A | Tesla Model S, Tesla Model X |
| 3 × 32 A | Renault Zoe, Smart EQ ForFour |

Dominant in the Dutch EV-fleet are plug-in hybrid electric vehicles (PHEVs) in the 1 × 16 A vehicle category. This is largely due to tax exemptions that favored PHEVs on a similar level as BEVs. With PHEVs having close to 70% of the EV market share in the Netherlands (January 2019, [33]), the 1 × 16 A category is dominant in this analysis (see Table 4). Due to an electric Car2Go sharing platform in the city of Amsterdam (300 Smart EVs) the percentage of 3 × 32 A chargers is likely slightly higher than in the Netherlands as a whole. Other than that, the fleet composition found in the Amsterdam pilot is likely representative for the rest of the Netherlands.

The effective power a charging EV will experience also depends on the occupancy of the station. If two cars are simultaneously connected on both sockets, the current of the charging station has to be shared when the total number of connected phases exceeds 3 (for example a 1 × 16 A and 3 × 16 A model). The software can provide full power to both sockets if only three phases or less are charged.

This leads to more than 100 possible scenarios, since the specific combination of EV models determines if the current is shared. The impact of the smart charging profile will be different for these scenarios. In some scenarios a vehicle will not be able to profit from higher charging currents at all given that the EV model itself has limitations in current levels (EVs with a maximum of 16 A will generally not profit from the higher current provided by Flexpower). In some scenarios EVs will only partially profit from higher Flexpower currents, depending on the configuration of two connected vehicles and their limitations (on Flexpower stations there is 40% more current available to share during double occupancy, while the benefit might be 100% during single occupancy). This makes it important to estimate which EV model was charging in each session. This estimation is based on the RFID, which is assumed to be uniquely associated with the vehicle itself. For each RFID the highest power that was encountered at any transaction is selected as well as the largest amount of energy charged in a single session. If the RFID has sufficient charging sessions in our data set, we can assume the car at some moment in time has charged at (or close to) its maximum power and has charged an amount of energy close to its battery capacity. The results are presented in Figure 4.

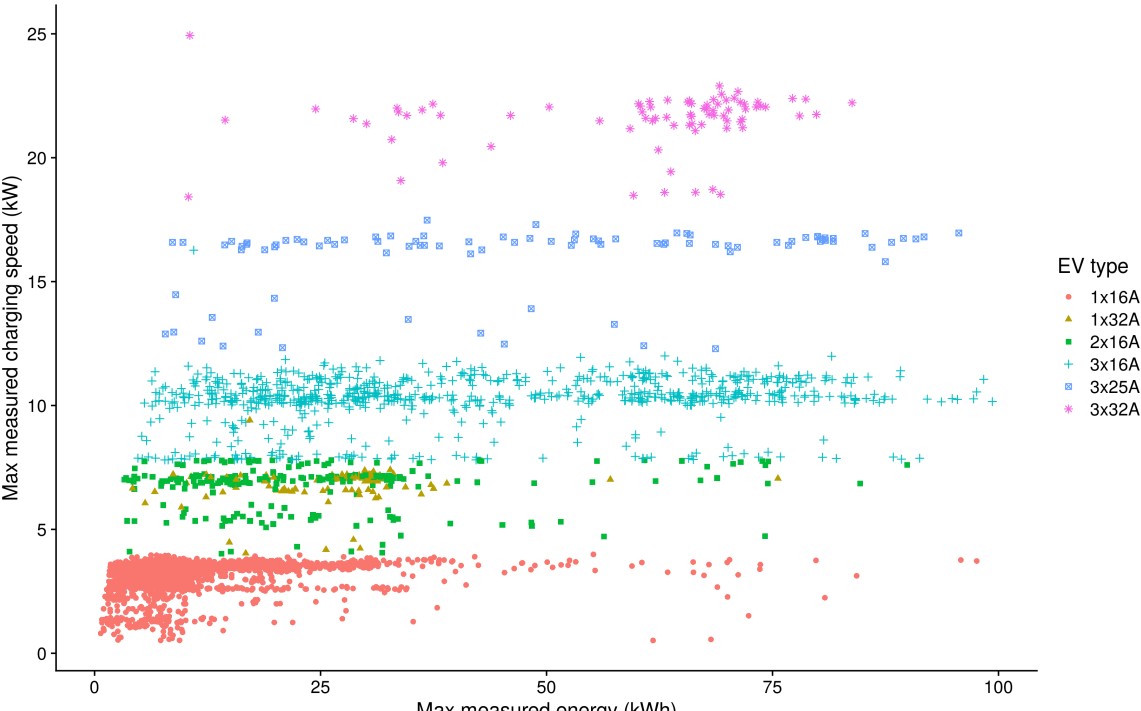

**Figure 4.** Scatter plot showing the highest measured charging speed versus the largest amount of energy in a single session for each RFID. The coloring and shapes indicate in which category the RFIDs are classified. Some RFIDs had insufficient data to be accurately classified into one of the categories because they had not visited a Flexpower station or only during conditions where the vehicle was not the limiting factor. These RFIDs are not shown in this graph.

In Figure 4 we can clearly identify different plateaus in the charging speed (vertical axis). The energy (horizontal axis) shows a more continuous distribution, which is to be expected since batteries will often not be completely depleted before charging. It can be noticed that there are hardly any cases where a vehicle has a large battery capacity but very low charging speed, which would indeed be an unfortunate combination.

**Table 4.** The distribution of RFIDs, sessions and total energy over the different vehicle categories for Flexpower and reference sessions combined. This shows that $1 \times 16$ A is the predominant category in the current market, but also that the energy per RFID (and per session) is much higher for the $3 \times 32$ A, suggesting more intensive usage.

| Vehicle Category | Number of Sessions | Number of RFIDs | Total Energy (MWh) |
|---|---|---|---|
| $1 \times 16$ A | 22,187 (71.6%) | 4437 (69.5%) | 132 (46.5%) |
| $2 \times 16$ A | 884 (2.9%) | 140 (2.2%) | 10 (3.5%) |
| $1 \times 32$ A | 1066 (3.4%) | 110 (1.7%) | 12 (4.2%) |
| $3 \times 16$ A | 2609 (8.4%) | 508 (8.0%) | 56 (19.6%) |
| $3 \times 25$ A | 618 (2.0%) | 84 (1.3%) | 14 (5.0%) |
| $3 \times 32$ A | 1090 (3.5%) | 60 (0.9%) | 35 (12.3%) |
| unknown | 2513 (8.1%) | 1044 (16.4%) | 25 (8.8%) |

The plateaus in the power distribution are the direct result of the charging characteristics of the EV model in terms of number of phases and maximum charging current. For example, the 17 kW plateau is associated with $3 \times 25$ A charging ($230 \text{ V} \times 3 \times 25 \text{ A} = 17.3$ kW), which is a characteristic of the Tesla Model S. Similarly, 11 kW is associated with $3 \times 16$ A charging, 3.7 kW corresponds to $1 \times 16$ A etc. The 7.4 kW plateau required an extra analysis step because it can be reached via $1 \times 32$ A charging, but also via $2 \times 16$ A. To make this distinction, the charging sessions on reference stations (which are limited to 16 A by the presence of the fuse) were compared to sessions on Flexpower stations. RFIDs that were able to charge 7.4 kW on both reference and Flexpower were classified as $2 \times 16$ A, while RFIDs that could charge only 3.7 kW on Reference and 7.4 kW on Flexpower were classified as $1 \times 32$ A. The classification allows us to give more focused insights on the impact of smart charging. An overview of the number of vehicles (RFIDs), sessions and the total energy per category is given in Table 4.

## 3. Results

In the following section we present the results of the impact the Flexpower profile has on actual charging behavior. Results are presented of the KPIs (i) charging speed, (ii) charged volume (in kWh) and (iii) positively/negatively affected sessions on Flexpower charging stations compared to the reference stations.

### 3.1. Charging Speed

The smart charging profile of Flexpower imposes a limit on the charging current in the evening (between 17:00 and 20:00) and morning time (07:00–08:30) which directly affects the charging speed (in comparison to the reference charging stations where the current remained 25 A). Flexpower provided a higher current limit of 35 A (compared to 25 A on reference stations) during the remaining hours. To investigate the impact on the effective charging speed of the six different vehicle categories as defined in Section 2.5, we plot the average power of all charging vehicles on the Flexpower and reference stations as a function of time of day. Since the time-dependent profile is the same on all Flexpower stations, the results for all stations can be aggregated.

Figure 5 shows the charging dynamics for the different vehicle categories. The colored lines represent the average power for each 15 min interval averaged over all active sessions during working days in the eight-month period. The graph is subdivided into six facets because the charging characteristics of different EV models on the market are very different.

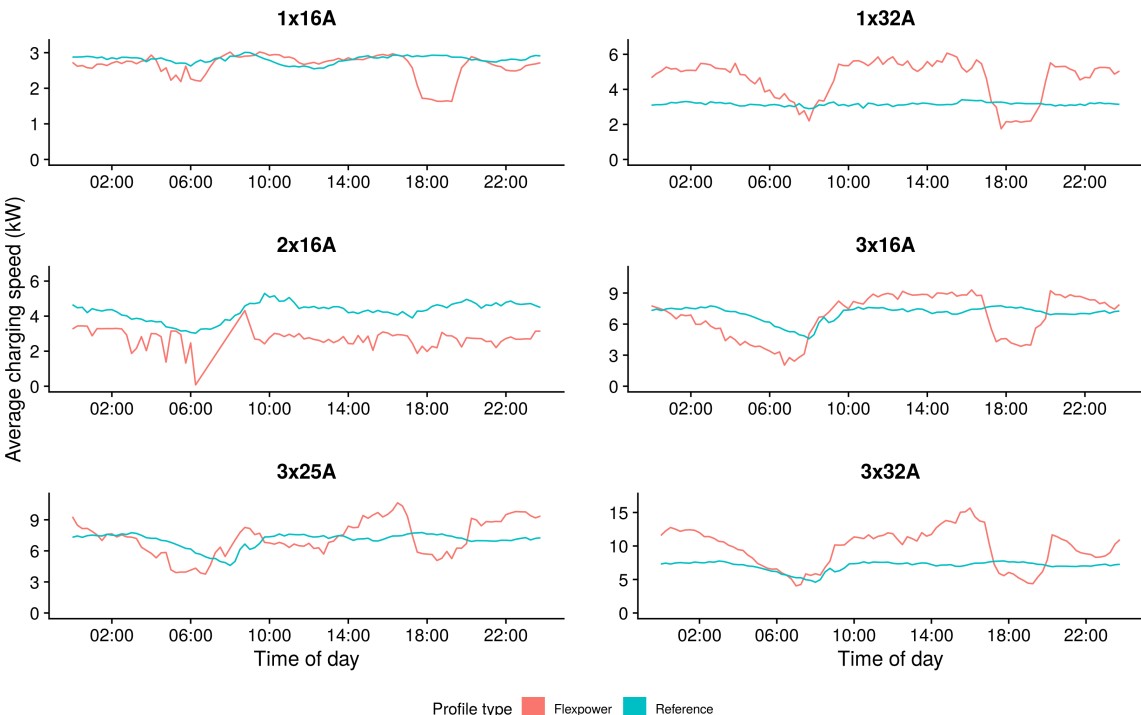

**Figure 5.** Grid showing the average power during the day for the different vehicle categories during charging. The resolution of the graph is 15 min, which is limited by the resolution of the metervalue data.

The blue line is calculated from sessions on reference stations, which always have a limit of 25 A for both sockets combined and have 16 A fuses on the individual sockets. It is interesting to note that the reference stations offer the same condition all day but nevertheless the charging speed fluctuates over time, especially for the categories charging on more than one phase. This shows that there are other factors besides the charging station characteristics that determine the effective power.

The red line shows the average power on Flexpower stations that have a time-dependent current limit. All categories show a reduction in power during the evening hours (17:00–20:00) as a result of the lower available current. The rest of the dynamics differ between the vehicle categories.

The $1 \times 16$ A category is limited by the vehicle during the rest of the day, so no increase in power is visible, even if Flexpower can provide higher currents. The limiting factor is the vehicle itself. During evening peak hours, when Flexpower limits current, a reduction of 30–50% in charging speed can be seen for $1 \times 16$ A-vehicles at Flexpower stations compared to reference stations. The $1 \times 32$ A category shows the same reduction in power when the current is limited but shows an increase in charging speed with approximately a factor 2 during the rest of the day, which is in line with the theoretical expectations. These vehicles can now charge at 32 A instead of being limited to 16 A by the fuse in the reference stations. The average effective power for the 1-phase category is very stable and the value is close to what would be expected theoretically (1.4 kW for $1 \times 6$ A during evening peak hours, 3.7 kW for $1 \times 16$ A and 7.4 kW for $1 \times 32$ A).

The $3 \times 16$ A category is also limited by the vehicle during the day but still shows some advantage of Flexpower. The $3 \times 25$ A and $3 \times 32$ A categories show a similar result to the $3 \times 16$, but the charging speed is higher during the day, where the $3 \times 32$ A vehicles have highest advantage. However, the increase in power is far from the factor 2 that was seen for the $1 \times 32$ A category. In general, the 3-phase categories show more erratic behavior and the effective power is far below the theoretical expectation (11 kW for $3 \times 16$ A, 17 kW for $3 \times 25$ A and 22 kW for $3 \times 32$ A). The charging process is apparently more complex for these vehicles or there is more variation between the vehicles causing them to charge at lower power than their theoretical maximum. One known effect is that 3-phase

vehicles are cut in power when other vehicles are connected and charging simultaneously on the same charging station (33% of the sessions, but this varies over the day). This occurs because the total current limit must be shared over more than three phases. In this case the station will offer both sockets half of the available current, which is 12.5 A for reference stations. This current limit sharing effect also explains the advantage 3-phase vehicles have on Flexpower stations during the day even though they are internally limited to 16 A: There is 35 A to share during double occupancy instead of 25 A. The current limit sharing effect can also occur for 1-phase vehicles, but such a vehicle most often shares the station with another 1-phase vehicle during double occupancy (1-phase vehicles have the largest current market share) and therefore rarely exceeds the total limit of three phases for the whole station which causes the current limit to be shared.

The 2 × 16 A category shows very counter-intuitive results. The behavior is expected to be similar to the 1 × 16 A and 3 × 16 A categories, but Flexpower gives about 50% less effective power during the whole day for this category. This cannot be explained using electrical engineering and must be the result of a fault in the system. It seems Flexpower only allows 2 × 16 A vehicles to use one phase for charging. The fact that there is very little data in this category makes further interpretation difficult.

All the categories in Figure 5 show that the charging speed tends to decrease during night times. This is caused by vehicles reaching high state-of-charge leading its battery management system to reduce charging speed to trickle charging. Overall this leads to a declining trend during the night as more vehicles are approximating fully charged batteries.

All in all, Figure 5 shows that the effective power is different than the maximum theoretical power for the categories. Known influencing factors include double occupation of a charging station (the current limit is shared between the vehicles), different charging behavior depending on the battery state-of-charge (constant voltage, constant current), user specific configurations in the vehicle, external conditions like temperature. It is not possible to quantify all these effects with the available data, but because these factors are present equally for both the reference stations as the Flexpower stations, we can attribute the difference in effective power to the time-dependent profiles, therefore we are effectively measuring the impact of the smart charging strategy on the actual charging speed.

### 3.2. Volume of Energy Transfer

The results in Section 3.1 show that the average power is higher on Flexpower charging stations. However, this does not necessarily lead to higher energy volumes. For all sessions that result in a fully charged battery, the amount of energy will be the same, just charged in a smaller amount of time. For sessions that were constantly charging while being connected, the energy transfer could both have been lower (during peak times) as well as higher (during off-peak times) as a result of Flexpower.

Figure 6 shows that during the evening peak hours (17:00–20:00) the Flexpower sockets give up to 50% less energy than the reference stations and that this is compensated directly after 20:00 when the current limitations are lifted. During the rest of the day the Flexpower stations offer a higher current than the reference stations, but the average energy volume is comparable. This can be explained by the fact that when the full demand is already met by the reference stations, providing a higher supply only causes the demand to be met faster, but does not increase the demand. Because the charging sessions are spread out over the day, this faster charging is compensated by the increased amount of idle time giving the same net result. The delayed peak after 20:00 shows that the Flexpower infrastructure is capable of providing more energy if there is outstanding demand.

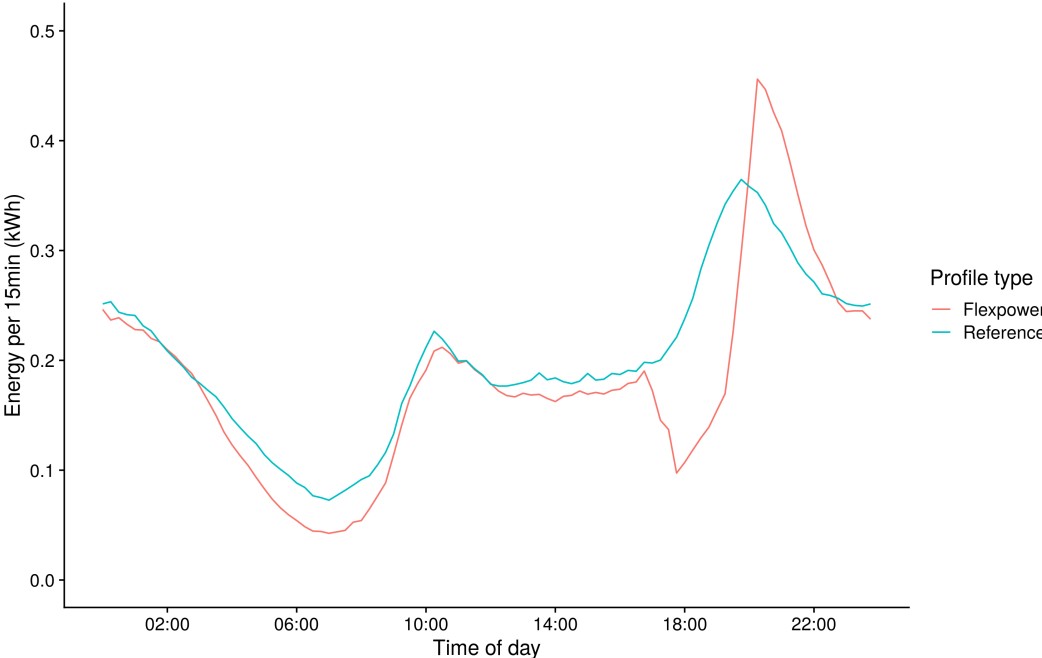

**Figure 6.** Line plot showing the average amount of charged energy per socket per 15 min interval. The value plotted in this figure is the total amount of energy charged during each 15 min interval divided by the total number of sockets and averaged over all weekdays in the 8-month test period. All charging sessions are considered and as such the figure does not distinguish between vehicle categories.

Of course, this result depends on the type of vehicle and the user behavior. If users would choose to pick up their car sooner because it has charged faster, a larger number of vehicles can be served by the same charging station and the total energy sales would be higher. Moreover, for EVs with large enough batteries to cover multiple trips, the higher charging speeds may convince users to connect their vehicles less often. This would also result in more vehicles being served by the same charging station and lead to higher energy volumes. Whether these kinds of behavioral changes in charging are present is object of further study.

### 3.3. Positively and Negatively Affected Sessions

An important indicator for smart charging in practice is the extent to which EV users are positively or negatively affected by providing a Flexpower profile compared to the current standard static charging profile. In terms of the vehicle categories as defined in Section 2.5, the 3-phase and >16 A categories represent full electric vehicles with higher charging current and larger battery capacity for which the new infrastructure should be beneficial. In comparison, the large population of PHEVs (mostly 1 × 16 A category) are likely to be disadvantaged by lower charging current in peak periods. Even though PHEVs do not solely rely on their battery, it is important to investigate the impact on their charging opportunities. The question then is, which share of sessions are positively and negatively affected by the Flexpower profile. This can be operationalized in the amount of energy a vehicle can charge during connection for the transactions at both Flexpower as reference stations. However, since the amount of energy is dependent on for instance battery size of the EV and/or the state-of-charge (SOC) of the batteries we prefer to analyze this indicator by looking at the average power per charging session. The average power is directly proportional to the amount of energy charged and is insensitive to effects of large batteries and SOC.

For many charging sessions, especially those taking place overnight (43% of all sessions), the user is not affected by the changes introduced by the Flexpower profile. The car reaches a full state-of-charge regardless of the type of profile because the connection time is (much) longer than what is necessary

to fully charge the battery. We therefore assume that the user is unaffected for all sessions that stop charging during the connection time. There is a small group of sessions where the difference in charging speed between Flexpower and reference would exactly make the difference between a full battery or not, but we can neglect these cases because they are rare, difficult to pinpoint and not very relevant because especially a low state-of-charge is critical for EV drivers, and these cases all concern a battery that is almost fully charged. Sessions which have completed charging can be recognised in the data because the transfer of energy goes to zero at some moment during the session and these are flagged as being completed.

The histograms in Figure 7 show how the power distributions vary between reference and Flexpower stations for each vehicle category. For some vehicle categories there are large differences between the distributions, but in all cases the power distributions have at least some overlap, making it difficult to pinpoint the exact share of positively and negatively affected users. In order to highlight the differences rather than the similarities between the power distributions, we subtract the amount of reference sessions in each bin of the histogram from the amount of sessions with the same average power observed on Flexpower stations. As such, the similarities in the distributions between Flexpower and reference stations, which we interpret as unaffected sessions, were removed, leaving the differences in distributions between the two types of stations, which we interpret as affected sessions. Figure 8 shows an example of the difference between the power distributions for the $1 \times 32$ A vehicle category. When the difference is larger than 0, there are more sessions on Flexpower stations which have experienced the corresponding average power. Similarly, when the difference is less than 0 there were more sessions on reference stations with this average power. As such, the figure provides a clear overview of the shifts in power distributions as a result of Flexpower for a particular vehicle category.

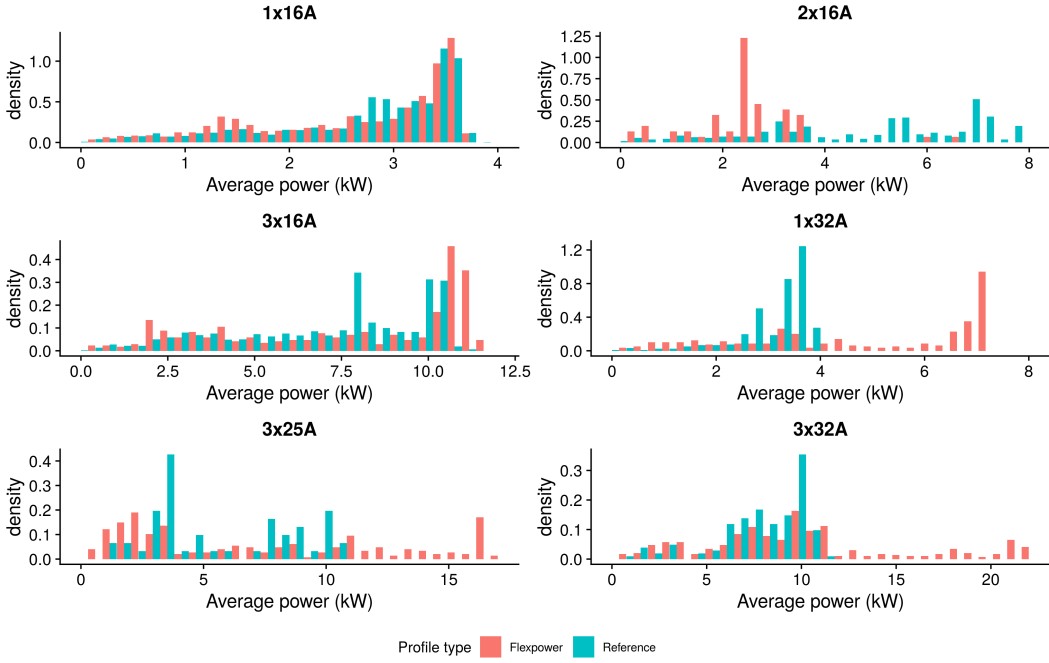

**Figure 7.** Histogram per vehicle category showing the distribution of average charging speed per session for Flexpower and reference stations. The average is calculated for the whole session, so periods of slower charging during the evening hours can be compensated during the preceding or following hours. Only sessions that have not finished charging upon disconnection are shown (47.3% of the sessions).

Zooming in, two zones can be identified where the difference is larger than 0 corresponding to two shifts in power, one on the low-power side and one on the high-power side. The area in between where the difference is negative represents the original power values that are now underrepresented

in the Flexpower distribution. We interpret the zone on the low end of the power axis (colored red) as the negatively affected sessions, as they represent the increased amount of sessions with a lower power compared to reference stations. The lower power is the result of lower available current during the limitation hours. The zone on the high end of the power axis (colored green) is interpreted as the positively affected sessions, as they represent the increased amount of sessions with a higher power. This is the result of higher current levels available during the rest of the day. This method was evaluated for all vehicle categories leading to the results in Table 5.

The results in Table 5 depend strongly on the charging characteristics of the vehicle categories. $1 \times 16$ A chargers only use a single phase and are limited to 16 A. This means they cannot profit from higher current levels in the Flexpower profile, even when they charge simultaneously with another vehicle on the same station since both phases do not influence each other. These vehicles do suffer from the current limitations during the evening hours, leading to lower charging speed for corresponding sessions (this is also visible in Figure 7). Only 9% of all sessions by $1 \times 16$ A vehicles are negatively affected; 90% was not affected.

The $2 \times 16$ A vehicles show 64% negative impact, which is also visible in Figure 7. It seems there is a technical problem for these vehicles on Flexpower stations causing them to only charge on a single phase. There is no theoretical explanation for this behavior; in theory similar percentages as the $1 \times 16$ A category should be expected. The high percentage of negatively affected sessions should thus be explained by technical difficulties rather than the Flexpower itself. It should be noted that the market share of this category is very low (<2%).

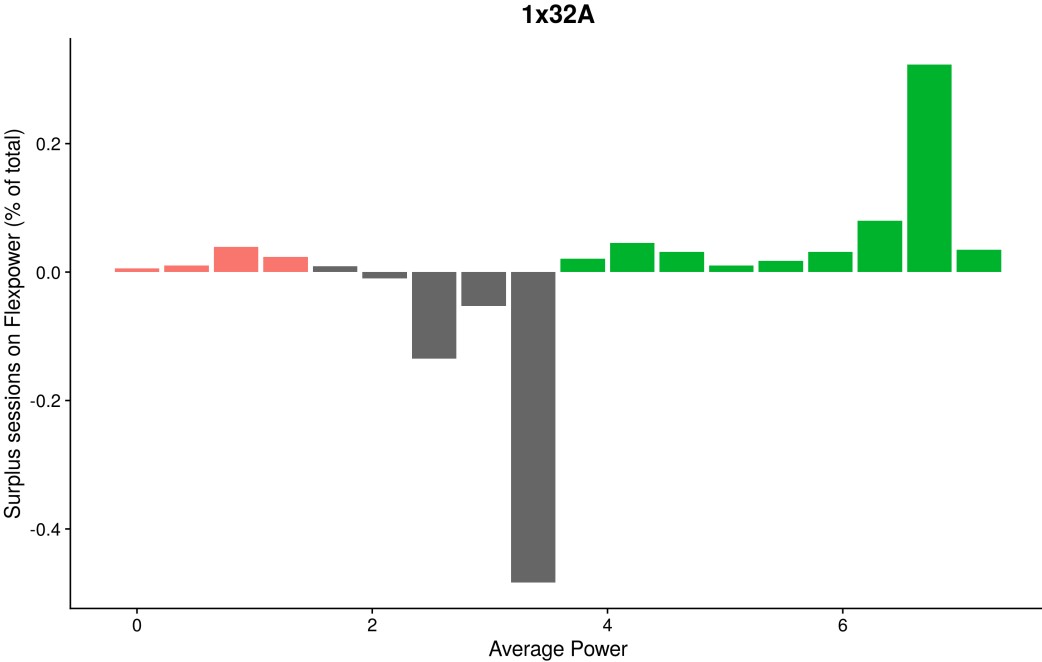

**Figure 8.** The difference between the power distributions on Flexpower and reference stations for the electric vehicle (EV) category $1 \times 32$ A. The y-axis has been normalized such that the value of a bar in the histogram corresponds to the fraction of the total number of sessions. The sum of all positive bars (which by definition is equal to the sum of all negative bars) corresponds to the percentage of sessions that is affected by Flexpower.

**Table 5.** Table showing which percentage of charging sessions was influenced by Flexpower and how. The numbers represent the proportion of sessions in the red and green areas of Figure 8. The numbers only reflect the sessions that were not completed at the moment of disconnection. All sessions that are completed are assumed to be unaffected.

| Vehicle Category | Negatively Affected | Not Affected | Positively Affected | Percentage of Sessions that Have Completed Charging |
|---|---|---|---|---|
| 1 × 16 A | 9% | 90% | 1% | 53.5% |
| 2 × 16 A | 64% | 35% | 1% | 49.1% |
| 3 × 16 A | 13% | 62% | 25% | 56.4% |
| 1 × 32 A | 7% | 46% | 47% | 55.4% |
| 3 × 25 A | 20% | 46% | 34% | 62.0% |
| 3 × 32 A | 8% | 59% | 33% | 55.5% |

The 3 × 16 A vehicles are limited to 16 A, just like the 1 × 16 A vehicles, but charge over three phases so they have to share the current during double occupancy. In this scenario the Flexpower profile gives an advantage because both sockets will still be able to provide 16 A, while on reference stations the current is split to 12.5 A each. This effect is clearly visible in Figure 7, where the peak at 8 kW on reference stations (3 × 12 A) shifts to 11 kW (3 × 16 A) on Flexpower stations. These vehicles still suffer from lower currents during evening hours, giving rise to a both a group of positively (25%) and negatively (13%) influenced sessions. The result of having both positively and negatively affected users also applies to the 3 × 25 A category, although the positively affected group is larger than the 3 × 16 A category (34%) because of the higher current that can be utilized by the vehicles.

The 1 × 32 A category experiences the largest advantage from Flexpower since these vehicles can charge at double the current and most of the time do not have to share the current during double occupancy. The negative impact of the current limitations is still present, but the positive impact is the highest value found in this study at almost 50% of the sessions. The 3 × 32 A category also has a large advantage (33%), but has a lower fraction of sessions that actually charge at double power. This can be explained by the fact that the vehicles cannot access the full current at double occupancy because they charge over three phases.

Overall 91% of the sessions are unaffected (including the completed sessions, the contribution of which are stated in Table 5), 4% are positively affected and 5% are negatively affected. These numbers are dominated by the 1 × 16 A category which has by far the highest share in the current EV market. Moreover, the fact that all completed sessions are unaffected has a large impact on these numbers. When we look at the car of the future (only categories with >16 A or 3-phases), the numbers become 14% positively affected, 5% negatively affected and 81% unaffected. The percentage of negatively affected sessions does not change, but there is a much higher percentage of positively affected sessions. Most of the sessions are still unaffected, which is to be expected (for example almost all overnight sessions are unaffected). Smart charging still has benefits in these cases because the timing of the charging can be optimized (e.g., delayed until late at night) without impacting the user.

## 4. Discussion

This paper presents the results of applying a time-dependent current profile, including a current limit during peak hours and a current surplus during off-peak hours. A pilot was evaluated in which more than 40,000 charging sessions were recorded by real users in the city of Amsterdam over a period of eight months. The results show a large difference between the theoretical charging limit and the practical speed that is realized. For example, for the 1 × 16 A vehicles the actual charging speed is stable around 3 kW (Figure 5), while the theoretical limit for 1 × 16 A is 3.7 kW, a difference of about 20%. This discrepancy can be found for all categories and is an important insight to help make policy and models more realistic. This difference between theoretical limit and the charging speed in practice

arises from the sum of many factors, some associated with the vehicle and some associated with the charging station and the grid. It is difficult to say to what extent this result applies to different cities and countries as the local circumstances may differ significantly for public charging infrastructures in terms of connection types, vehicle fleet composition and occupancy rates.

The paper also shows that it is possible to limit the energy consumption within a 3 h time window without large impact on EV drivers, proving that smart charging is a viable solution for balancing the load on the electricity grid particularly if higher current levels are provided during off-peak hours. Since the consumer impact is positive specifically for more advanced vehicles (in terms of current limit and number of phases), the potential for applying this measure will increase further as the fleet composition will move to vehicles that charge faster (higher current levels and/or more phases).

The paper shows that the current implementation leads to a rebound peak. This is not necessarily a problem since the load of all other connections on the grid (such as households) may have reduced sufficiently by the time current limitations are lifted, but it is not necessary and could be avoided by applying a more gradual increase in the current limit after peak hours.

The ambitious renewable energy targets in Amsterdam and corresponding growth in EV market share will make a smart charging strategy unavoidable. The application of a time dependent profile should be implemented first in areas where there are many charging stations on a single transformer, which is likely the first weak spot in the grid infrastructure, or on known grid networks with limited capacity left. A follow-up study is currently carried out to make the connection between grid measurements in the low and medium voltage grids and the EV charging data.

A dynamic approach would offer an optimal consumer service and more flexibility than the current static time-dependent profile. For example, a limit could be imposed on an area serviced by a single transformer, and only when the sum of all sessions passes this threshold would the charging speed be regulated accordingly. This avoids unnecessary speed reduction, but requires real time communication between the CPO and the sensor infrastructure, which is supported by the OCPP protocol, but is not currently in place.

A second possible improvement on the current one-size-fits all implementation is to grant control to the consumer, possibly combined with a price incentive. The user knows their intentions best and it seems superfluous to train complicated and incomplete predictive models instead of just involving the consumer. Depending on the estimated connection time and amount of required energy an optimal profile can be calculated. This avoids the effect of impacting vulnerable charging sessions but reaches almost the same overall effect. An experiment has been done with a 'stop button' in Gelderland [34] and preliminary results show that it is not misused to circumvent the charging limitations.

Lastly, the results in this paper show that in the current situation the possibility of increasing charging volumes during the day is limited by the level of demand and technical limitations of most of the electric vehicles currently on the market. If the goal of better overlap of EV charging with solar power generation is to be realized, consumers need more incentives to charge during the day and increase the percentage of 'green' charging.

## 5. Conclusions

A time-dependent current limit was deployed on 39 public charging stations in the city of Amsterdam where the current was reduced during the peak hours of household energy consumption (07:00–08:30 and 17:00–20:00) but was increased during the rest of the day. By alternating a lower current during peak hours with a current surplus during off-peak hours we were successfully able to suppress the load of EV charging on the grid during a designated time window with minimal consumer impact.

In total 91% of all sessions were not affected in terms of amount of charged energy, with only 5% of all sessions being negatively affected and 4% being positively affected. The group of consumers with most negative impact are PHEV models which have a good alternative in the form of their internal combustion engine. Advanced BEV models with higher charging speeds can profit from the Flexpower

profile due to the surplus current during off-peak hours. With a PHEV-dominated fleet composition in the Netherlands (and in this pilot), the results of Flexpower may prove to be even more positive for countries that have a higher share of BEVs compared to PHEVs.

The results of this study show how effective charging speed in real-world measurements is about 20% lower than the theoretical value based on the current limit and standard voltage. Moreover, this study underlines the importance of taking double-occupancy and differences in charging characteristics of EVs into account in order to make more realistic models and predictions of charging behavior and effects of applying smart charging.

Even though it is does not address local differences in the household grid load and EV charging behavior, the current one-size-fits-all Flexpower implementation is already suitable for large-scale roll out. As such, it is recommendable to consider for policy makers in cities with a high take-off of EVs. Applying smart charging profiles such as Flexpower enables cities to implement large scale introduction of EVs while reducing concerns on grid impact with minimal negative effects on the user. In comparison to other smart charging strategies the static Flexpower is particularly suited for addressing the issue of grid impact, rather than reducing charging costs (by trading on energy markets). More dynamic smart charging strategies may be used to tailor charging current levels to accommodate matching with renewable energy generation and match with (seasonal) peaks in energy usage, something considered in follow-up work to this pilot. The pilot was carried out on public charging stations, but the results may also apply on reducing grid impacts of residential charging, although policy makers currently have little influence on charging on local level.

**Author Contributions:** R.v.d.H. and F.G. conceived the presented idea and supervised the project. P.C.B. carried out the analyses with the help of A.B. and G.L. N.P. advised on the experimental design and helped interpret the results. All authors have read and agreed to the published version of the manuscript.

**Funding:** This study was made possible with the support of the Interreg North Sea Region Programme through the project SEEV4-city.

**Acknowledgments:** The authors would like to thank all partners participating in the Flexpower project including Vattenfall, Qirion, ElaadNL and the City of Amsterdam.

**Conflicts of Interest:** The authors declare no conflict of interest.

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
