# Peer review of "Impact of Smart Charging for Consumers in a Real World Pilotâ€"

_wevj, doi:10.3390/wevj11010021_

Round 1
Reviewer 1 Report
The authors provide an extensive experimental study on the impact of implementing smart charging profiles, through time-of-day charging limitations, in the city of Amsterdam. The study considers the use of public charging stations, on weekdays for a number of EV categories.
The results provide in depth evidence of the potential benefits of rolling out smart charging policies for public charging infrastructure that balances grid limitation with consumer needs.
Prior to publication the authors should address the following:
General comment
The authors mention grid impact on the title, however the grid characteristics for this case are not really approached in the manuscript, and the results are very much focused on the charging stations and the impact on EV users. No discussion on Amsterdam’s grid characteristics is added. This should be adjusted.
Introduction
P.1.L.19. It would be interesting to the EV growth, or EV market shares here.
Materials and methods
P.3.L.103-105 this can be a footnote, as it is not central to the paper, or perhaps just emphasize that the study focuses on weekday charging.
Data
P.4.L.118 add CHIEF acronym explanation
Conclusions
This section does not really provide any clear policy implications or guidance on how to act upon the results of this pilot results. Should smart charging be implemented? If so, how, why, where? How does Amsterdam benefit/could benefit from such a policy? This need attention from the authors.
P.15.L.383 remove one of the repeated “using” words
Discussion
This section read partly as conclusions and partly as discussion. The authors should reorganize the contents of this section accordingly. Move the discussion parts to a discussions section to be added before the conclusions, and the conclusion part to the conclusions section.
Reviewer 2 Report
The paper addresses an important topic: How could a smart charging profile benefit the electricity grid without affecting EV drivers negatively.
Here are some comments per section to support the authors in improving the quality of the paper:
Introduction: It would be good to link the chosen Flexpower profile with a clear positive impact on the electricity grid (and compare to other smart charging profiles). This is information that is missing throughout the manuscript. Especially, since the positive impact on EV drivers is not as large as one would have liked to see. The literature review, describes prior research only at high level without helping the reader obtain a good understanding of the actual problem and gap in the literature (which would be filled by the current study). It also handles mainly past simulation results whereas the current study presents experimental results. It is strongly advised to deepen the review but also to expand it to experimental work review. In line 64, it is mentioned that the presented study uses 40 pilots charging stations (how many are dual?) but later on in line 95 only 38 pilot stations are mentioned. This needs to be clarified. Materials and methods A clear and overall description of the methodology used is missing. The authors show a clear tendency to explain the methodology using words when in reality a simple flow chart and some relevant simple equations would help the reader understand much better. lines 103-105: is it realistic to exclude weekends and holidays? how big a variation did they introduced and why the authors deem it is ok to exclude it from the present study? Some visual/graphic justification is required. lines 167-169: 3 specific KPIs were selected. The authors are expected to explain why they think that the 3 KPIs are sufficient and what could be additional KPIs (and why they were left out). Figure 2 is not sufficient to provide the reader with a quantitative understanding of the difference between Flex Power and Reference. So is Figure 4. It is recommended that further analysis of the results is undertaken to provide more insight. Results The results presented lack conclusiveness. It is very difficult to understand how our knowledge on the benefits of the Flexpower profile was enriched compared to the introduction of the paper. Throughout the results (and generally the paper) the same notions of dependency on various factors are repeated. It would be beneficial to know e.g. which factor has a more dominant impact on the results or maybe via a parametric analysis e.g. show how the positively affected sessions could be increased. Questions to support the authors: How would this profile (FlexPower) compare to other smart charging profiles? Why this and not another profile? What are the benefits? Conclusion and Discussion The discussion remains mainly qualitative leaving the reader wondering about the effectiveness of the researched profile. There are no concrete recommendations by the authors. It is understood that further investigation is required but some first comparative conclusions should be drawn more clearly.Author Response
Please see the attachment

Reviewer 3 Report
Overall the manuscript presents one of the few analyses of smart charging completed on public charging infrastructure, in contrast to prior work which has primarily focused on residential charging. Clarifications on results interpretation as well as on the pilot design (and how it represents the broader context of EVs in the Netherlands and elsewhere) are needed, as well as more information on the specific contributions to the literature and how results support or oppose prior studies. Specific comments are provided below:
Line 20-21: Not clear what is meant by 15-20% additional electricity? Over the course of the year, during peak times? At what level of vehicle adoption? Is this specific to the Netherlands? Please clarify and provide a citation. Line 48: citation needed, this statement is also very context dependent Lines 54-57: Please provide citations of example studies that show the benefits of more realistic assumptions. In other words, what is the implication of these simplified assumptions in modelling studies? Lines 59-60: Please briefly summarize what these “early results” are from prior smart charging pilots in the literature, so that the results of this pilot can be interpreted in context of the literature. Lines 60-62: It would be important to mention here any different characteristics between residential and public charging that would affect the feasibility and behavior of smart charging programs. Why is it necessary to pilot smart charging in public chargers if it has already been piloted at the residential level? What share of charging occurs at residential locations compared to public chargers in the Netherlands/Amsterdam? (For example, in the United States, public chargers are a much smaller share of charging infrastructure compared to residential charging.) With increased adoption of electric vehicles, is charging expected to increase in public locations? Line 66: Please include a citation or additional description of Flexpower. Has it been used elsewhere for smart charging or was it a software created specifically for this pilot? Which of the 3 smart charging optimization objectives mentioned above are targeted by Flexpower? Lines 89-92: Given that the charging current is altered to set levels at set times, it appears that the pilot is closer to a “time-of-use” static rate than a smart charging program, which usually dynamically controls charging times or rates given real-time conditions on the grid or market prices. Please elaborate on the limitations of this pilot design, namely that the peak and off-peak times are pre-determined and static Lines 93-94: What is the current share of solar and wind generation in the Netherlands? Line 95: the introduction mentioned 40 charging stations and here it is 38. Please make sure it is consistent. Line 97-99: Some additional information about the pilot design and big-picture statistics about charging behavior in the study area would be helpful. How were the smart charging station locations chosen? Were drivers informed in any way that the charging rate would be changed through the pilot, or did they unknowingly plug into smart charging stations? How many total public charging stations are there in Amsterdam, and what share of charging typically occurs at public chargers compared to residential or work chargers? What is the average utilization rate of public chargers? Line 131: I think you mean described below, instead of above. Better to include actual section numbers to cross-reference. Line 164-165: Does KPI stand for Key Performance Indicator? Please spell out abbreviation on first reference. How did you choose these particular metrics? Did you speak with specific stakeholders (drivers, electric utilities, grid operators, etc.) or are these commonly used metrics from the literature or other smart charging pilots? Line 175: Please clarify what is meant by pilot stations relating to “particular geographic areas” and how the locations were chosen. Lines 181-182: Please include the SD, max and min of the metrics in Table 2 so it provides a better comparison of the distribution of the data, especially because the caption says the data is sensitive to some outliers. How would the results change if those outliers mentioned (extremely long connection of 3.5 weeks) were removed from the data set? Line 185: Can you provide some data in Table 3 on the vehicle adoption rates of these different models in Amsterdam or the Netherlands to provide some context and show how representative the sample of vehicles in the pilot are of the broader vehicle population in the country. Are these all fully-battery electric models or do they include plug-in hybrid models as well? If both PHEVs and BEVs are included in the pilot, both types of models should be listed in this table. Line 222: Does table 4 include both reference and smart charging sessions? Please clarify and/or separate the columns between smart charging and reference sessions. Line 302: broken reference to section number Line 303-304: Even if PHEVs have smaller battery capacity, I’m not sure they would be so clearly disadvantaged compared to BEVs with larger battery range since PHEVs aren’t solely reliant on the electric motor. Also, here it is mentioned that the PHEVs are mostly 1x16A, but in table 3, the only model on the market shown is the Nissan Leaf which is a BEV. Line 313: What share of charging session occur overnight? Line 330: It appears there are two types charging sessions that are being classified as “unaffected sessions”: 1. Smart charging sessions when the vehicle reaches its full SOC by the end of the charging session, and 2. Charging sessions that reach the same average power level per session between the reference and smart charging sessions. Unless these two types of sessions are being treated equally in calculating the performance metrics, it is quite confusing to classify these two cases with the same name since the 2nd type of session may not reach its full SOC, and just has the same power level as the reference set. Please use different terminology when describing the two types of sessions for clarity. Lines 337-343: Figure 7 and its interpretation is not very intuitive (and needs units on the x-axis also). Please write a more clear explanation of why the area to the left of negative bars represents “negatively affected” charging sessions. Also in Table 5, I think the caption should refer to red and green bars rather than red and blue. Line 372: grammar: change “is” to “are unaffected” Line 374: It is unclear from this sentence whether the 91% of unaffected sessions include the completed sessions or not? Please clarify. Lines 384-385: How do the results of this pilot compare to the findings from prior pilots in the literature? How is any difference in result explained by the use of public chargers instead of residential chargers? Or is there any other reason? Lines 405-406: It is unclear from the description of the pilot design if drivers are aware that the current is being reduced/increased at FLexpower stations. Is there any way to know if drivers who have been negatively affected have moved their vehicles to non-participating or reference stations? In other words, is there any evidence of a rebound effect? Lines 415-416: It should be noted here that the power levels are fluctuated to pre-determined levels at pre-determined times, which is a limitation of the pilot design. Line 418: Neither the conclusions or discussion section place the results in the context to prior literature. Please include some discussion of how these results compare to other smart charging pilots or time-of-use EV rates, and any limitations of the analysis. How much of these findings can be generalized beyond the context of Amsterdam/Netherlands? Line 435: What is meant by “radiation levels”? I think the authors may mean solar generation levels? Please change the word here (radiation level typically refers to radiation from nuclear materials!).Author Response
Please see the attachment

Reviewer 4 Report
The paper GRID AND CONSUMER IMPACT OF SMART CHARGING IN A REAL WORLD PILOT is an interesting case study on the impact of a smart charging infrastructure applied on 38 EV charging points. The finding are useful and interesting for further large scale studies and the work is well organised and well written. Some minor comments: Literature Review - When discussing about pro and cons of simulation the literature review should also take into account latest co-simulation infrastructure for demand response such as: https://doi.org/10.1109/SII.2019.8700423 https://doi.org/10.1016/j.softx.2019.03.003 It would be worth mentioning those. Line 85 - “Dutch household consumes on average around 1kW at peak moments.” - interesting number, it requires a back up reference (Italy is 3kW, Ireland is 6kW) Line 283 and Figure 5 - the figure shows a classic rebound effect after the demand side management event, it is well documented in the literature, what is missing is a simple expert evaluation of the implication of rebound effect in case of large scale deployment of such a time capped profile.Author Response
Please see the attachment

Round 2
Reviewer 1 Report
Thank you for submitting an improved version of the manuscript.
The new title reflects better the content and aim of the study, which I believe will benefit the work and the audience reading this paper.
The discussion now presented provides a better link between the results obtained and how they relate to possible applications.
The changes introduced in the conclusions have made it less clear and structured. The authors should consider revising the conclusion. Perhaps consider enhancing the structure of the conclusion section with a brief description of the study, the main results (already presented in the current version), and limitations of this approach, as well as future work (already presented in the current version).
Author Response
Thank you for your helpful feedback. Your remaining point is addressed in the attached word file.

Reviewer 2 Report
Thank you for addressing the comments.
Author Response
Thank you for your helpful feedback and we are glad to hear you are satisfied with the changes.
Reviewer 3 Report
The revised manuscript includes a much more clear case for the contribution to the literature to evaluate the 1) impact of specific simplified assumptions from typical simulation studies (double occupancy on charging speeds, and differences in charging speeds across EV models), and 2) smart charging with public chargers (as opposed to residential chargers which have more predictable behavior). In addition, clarifications on the set-up of the pilot study (including site selection, comparison with the control group, KPI selection) and the Amsterdam/Netherlands context were satisfactorily addressed. The interpretation of the results is also now better explained in this revised manuscript across the Results, Discussion, and Conclusion sections.
There remain a couple of minor typos and comments:
Line 23: Missing word/typo and still not clear that you mean annual average consumption as you explained your comments. Recommend rewording to: “The impact of electric mobility may amount up to an additional 15% of average annual electricity demand of households.”
Line 38: Spelling: Analyze
Line 100 and 116: The pilot isn’t an actual time-of-use rate (which is a financial incentive through time-dependent customer electricity prices), but rather is similar to the effect of a time-of-use rate. So I would rephrase the sentence to “As such, the smart charging profile has a similar effect as a 'time-of-use' EV electricity price with fixed off-peak and peak times, rather than a dynamic program where power levels are varied based on real-time conditions (e.g. such as market prices or grid congestion). I would also take out “time-of-use” from Line 116 and just refer to the program as a real-world case of a “static smart charging profile”.
Line 282: A bit confusing that there is a 70% PHEV market share in the Netherlands mentioned here, and a 7% EV market share mentioned for the Netherlands in line 20. Does the 7% market share only include BEVs?
Line 541: Change to “their intentions” from “his intentions”
Line 547: I don’t see a direct connection between the results of the paper and the statement here that “there is a higher energy supply than demand during the day.” More specific references to your results are needed to draw this conclusion. Is this because 43% of charging sessions occur at night?
Line 563: Grammar: Change “offers” to “enables” otherwise this sentence doesn’t make sense
Author Response
Thank you for your helpful feedback. Your remaining points are addressed in the attached Word file.
